# NCOA7 Regulates Growth and Metastasis of Clear Cell Renal Cell Carcinoma via MAPK/ERK Signaling Pathway

**DOI:** 10.3390/ijms241411584

**Published:** 2023-07-18

**Authors:** Jiayu Guo, Shuai Ke, Qi Chen, Jiangqiao Zhou, Jia Guo, Tao Qiu

**Affiliations:** 1Department of Organ Transplantation, Renmin Hospital of Wuhan University, Wuhan 430060, China; 2Department of Urology, Renmin Hospital of Wuhan University, Wuhan 430060, China

**Keywords:** clear cell renal cell carcinoma, MAPK pathway, epithelial–mesenchymal transformation (EMT), molecular target, NCOA7

## Abstract

NCOA7 is a nuclear receptor coactivator that is downregulated in a variety of cancers. However, the expression and prognostic significance of NCOA7 in clear cell renal cell carcinoma (ccRCC) remain unknown. The expression of NCOA7 in ccRCC tissues was analyzed using bioinformatics analysis, Western blotting, and immunohistochemistry. Kaplan–Meier analysis, the receiver operating characteristic (ROC) curve, and clinicopathological correlation analysis were used to assess the predictive power of NCOA7. Overexpression function tests were conducted in cells and mouse models to clarify the function and mechanism of NCOA7 in inhibiting the progression of ccRCC. NCOA7 expression was downregulated in all three subtypes of renal cell carcinoma, and only had significant prognostic value for patients with ccRCC. NCOA7 overexpression inhibited the proliferation, invasion, and metastasis of ccRCC cells in vivo and in vitro. Mechanistically, NCOA7 inhibited the MAPK/ERK pathway to regulate epithelial–mesenchymal transformation (EMT) and apoptosis, thereby inhibiting the progression of ccRCC. NCOA7 inhibits tumor growth and metastasis of ccRCC through the MAPK/ERK pathway, thus indicating its potential as a prognostic marker and therapeutic target for ccRCC.

## 1. Introduction

Malignant renal tumors are one of the most common cancers worldwide with increasing incidence, particularly in men and the elderly [1,2]. Renal cell carcinoma (RCC) comprises approximately 80–90% of malignant renal tumors in adults, with clear cell renal cell carcinoma (ccRCC) as its most frequent histological subtype, originating from the proximal tubular cells of the renal unit [2,3]. The current major treatment options for RCC include re-sectioning localized tumors and systemically treating metastatic disease with poor prognoses [4]. However, approximately 30% of patients with RCC have metastatic disease at the time of initial presentation, making treatment difficult and the prognosis challenging. Additionally, 30% of patients will still experience recurrence and metastases following the resection of the primary tumor [5]. Therefore, early diagnosis of RCC is important for early intervention. New diagnostic markers need to be discovered for the early diagnosis of RCC and to predict the prognosis of patients with RCC.

Dysregulation of the RAF/MEK/ERK (MAPK, mitogen-activated protein kinase) signaling pathway is closely associated with human tumorigenesis and has been shown to promote tumor proliferation, survival, invasion, metastasis, degradation of extracellular matrix, and angiogenesis [6,7,8]. The MAPK cascade is normally activated by the oncogene, RAS [9], and is dependent on the activity of downstream ERK to regulate a variety of important physiological functions. ERK can perform a range of different functions by phosphorylating various molecular substrates [7]. The mechanism of the MAPK/ERK1/2 signaling pathway, with regard to activation in RCC, has now been reported [10], which regulates the progression of RCC by promoting proliferation, migration, and invasion [11,12,13]. Thus, intervening in the MAPK/ERK1/2 pathway may be the most effective strategy to treat metastatic RCC and improve the prognosis for patients with advanced RCC.

Nuclear receptor coactivator 7 (NCOA7), also known as the ER-associated protein of 141 kDa (ERAP140), is a conserved tissue-specific nuclear receptor coactivator that has been shown to bind to estrogen receptor α (ERα) and to enhance the transcriptional activity of its interacting receptors [14]. ERα is closely linked to abnormal proliferation, inflammation, and malignant tumor development, and it influences the tumor response to targeted therapies [15]. NCOA7 has been shown to correlate with the prognosis of various tumors such as neuroblastoma, colon cancer, and breast cancer, among other tumors [16,17,18]; it plays an anticancer role by synergistically enhancing the promoter activity of the all-trans retinoic acid target gene with retinoic acid receptor α (RARα) [16]. Hence, NCOA7 holds significant importance for targeted immunotherapy in cancer. However, the prognostic effects and molecular biological processes of NCOA7 in clear cell renal cell carcinoma remain unexplored, and it remains unclear whether it is involved in ccRCC progression.

Bioinformatics technology has been widely used to explore the prognosis and development of tumors, with significant implications. In the present study, we found that NCOA7 was only significant for the prognosis of ccRCC, a subtype of RCC and particularly important for the prognosis of advanced ccRCC. Furthermore, we demonstrated that both the protein and mRNA expression levels of NCOA7 were downregulated in ccRCC tissues. The overexpression of NCOA7 suppressed the proliferation and metastasis of ccRCC cells in vitro and in vivo via inhibition of the MAP/ERK pathway. These findings contribute to improving our novel understanding of the diagnosis and treatment of advanced ccRCC.

## 2. Results

### 2.1. Analysis of the Expression Level, Prognostic Value, and Diagnosis of NCOA7 in Three Subtypes of RCC

To explore the expression level of NCOA7 in RCC using the TCGA database, as shown in Appendix A, NCOA7 expression level was downregulated in various types of cancers relative to the corresponding healthy tissues, including bladder carcinoma, breast cancer, renal cell cancer, and lung cancers. Furthermore, NCOA7 expression levels were significantly decreased in all three subtypes of RCC (clear cell renal cell carcinoma, ccRCC; kidney chromophobe, KICH; kidney renal papillary cell carcinoma, KIRP) compared with healthy paracancerous tissues of RCC (Figure 1A–F). However, the analysis of the prognostic value of the NCOA7 expression level on the three subtypes of RCC using Kaplan–Meier survival curves showed that the NCOA7 expression level was significantly associated with both OS and PFS in ccRCC patients only, independent of both KICH and KIRP. Moreover, patients with high NCOA7 expression in ccRCC had a better prognosis than those with a low expression (Figure 1G–L). Additionally, a ROC model was constructed using the TCGA database to assess the accuracy of the NCOA7 expression level in distinguishing ccRCC tissues from normal tissues. As shown in Figure 2A, the NCOA7 expression level had a good diagnostic value for ccRCC (>80%).

### 2.2. Correlation between the Expression Level of NCOA7 Regarding Clinicopathological Features and Prognosis

The relationship between the expression of NCOA7 and clinicopathological characteristics was investigated. As shown in Figure 2B–I, the expression of NCOA7 was closely correlated with age, gender, grade, and TNM stage. When the tumor was at a higher grading stage, the expression of NCOA7 was lower. Subsequently, the impact of high and low expressions of NCOA7 on different ages, stages, and grades on patient survival was compared separately. Across all age groups, patients with high NCOA7 expression exhibited better survival rates compared with those with low NCOA7 expression (*p* = 0.001, *p* = 0.009) (Appendix A). In terms of grade and stage, patients with high NCOA7 expression had better survival rates than those with low NCOA7 expression (*p* = 0.024, *p* = 0.007), but only when they were in the high-grade and advanced- stage (Appendix A), further suggesting that NCOA7 has a significant impact on the prognosis of patients with advanced disease. Additionally, the expression level of NCOA7 was negatively correlated with the tumor mutation burden (*p* < 0.001) (Appendix A).

### 2.3. The Expression of NCOA7 Is Downregulated in ccRCC and Cell Lines

To verify the expression levels of NCOA7 in ccRCC and normal tissues, the GSE53757, GSE66271, GSE6344, and GSE26574 datasets containing transcriptomic data were downloaded from the GEO database. After standardizing the microarray data, the data from NCOA7 expression were obtained; it was found that the mRNA levels of NCOA7 were significantly downregulated in ccRCC tissues compared with normal tissues (Figure 3A–D). Furthermore, analysis of NCOA7 protein expression levels was performed in ccRCC, and paired non-tumor tissues obtained from our hospital via immunohistochemistry and Western blot showed that NCOA7 protein expression levels were significantly reduced in ccRCC tissues compared with adjacent normal kidney tissues (Figure 3E,F).

Subsequently, the protein expression levels of NCOA7 in five different ccRCC cell lines were examined, and as shown in Figure 3G, 769-P and 786-O cells expressed lower levels of NCOA7 protein compared with the other ccRCC cell lines. Consequently, 769-P and 786-O cells were selected to facilitate the construction of stable NCOA7-expressing cell lines to explore the influence of NCOA7 overexpression on the malignant phenotype of ccRCC cells. Western blot assays showed that cell lines transfected with NCOA7 recombinant lentiviral plasmids expressed significantly more NCOA7 protein compared with the empty vector (vector) of 769-P and 786-O cell lines, thus confirming the success of overexpression (Figure 3H).

### 2.4. The Overexpression of NCOA7 Inhibits the Proliferation, Migration, and Invasion of 769-P and 786-O Cells In Vitro

To examine the influence of NCOA7 overexpression on the proliferation, migration, and invasion ability of ccRCC cells. Firstly, cell viability was measured using a CCK8 assay and the results showed that NCOA7 overexpression significantly inhibited the viability of 769-P and 786-O cells compared with the vector group (*p* < 0.01) (Figure 4A,B). The same result was further confirmed using a colony formation assay (Figure 4C,D). To further explore the impact of NCOA7 overexpression on the proliferation capacity of ccRCC cells, we examined the expression of PCNA via immunofluorescence staining. As shown in Figure 4E, compared with the vector group, the expression of PCNA in the NCOA7 overexpression group was significantly reduced, thus indicating that the proliferation capacity of ccRCC cells was significantly reduced. In addition, the results of the wound healing assay showed that the overexpression of NCOA7 significantly reduced the migration rate of 769-P and 786-O cells compared with the vector group (*p* < 0.001) (Figure 5A,B). Transwell invasion assays also showed that the invasive ability of ccRCC cells was significantly inhibited by NCOA7 overexpression (Figure 5C,D).

To characterize the relationship between NCOA7 and epithelial-mesenchymal transition (EMT), the epithelial marker, E-cadherin, and the mesenchymal marker, Vimentin, were labeled using dual fluorescence, and they were observed using fluorescence microscopy. NCOA7 overexpression resulted in a significant upregulation of E-cadherin expression and a significant downregulation of Vimentin expression in ccRCC cells (Figure 5E). This result was consistent with the Western blot assay, in which the expression of the E-cadherin protein was upregulated, and the expression of N-cadherin and Vimentin proteins were downregulated in ccRCC cells overexpressing NCOA7 (Figure 5F). In summary, our experiments have demonstrated that NCOA7 inhibits the proliferation, migration, and invasion of ccRCC cells in vitro.

### 2.5. The Effect of NCOA7 Overexpression on Cycle and Apoptosis in 769-P and 786-O Cells

To validate the effect of NCOA7 on ccRCC cell cycles and apoptosis, flow cytometry was used for detection purposes. The aberrant activity of cell cycle proteins usually plays an important role in the development of ccRCC and it acts a driver of tumorigenesis [19]. Analysis was performed using flow cytometry after the overexpression of NCOA7 in 769-P and 786-O, and it revealed an increase in the percentage of the G1 phase in ccRCC cells overexpressing NCOA7 (Figure 6A,B). Furthermore, it demonstrated a reduction in the protein expression of the G1/S specific cyclin Cyclin D1, compared with the vector group (Figure 6E). Additionally, the results of apoptosis detection via flow cytometry showed that NCOA7 overexpression increased the rate of apoptosis in 769-P (*p* < 0.001) and 786-O (*p* < 0.001) cells (Figure 6C,D). The Western blot assay showed that NCOA7 overexpression increased the expression of apoptotic proteins Bax and cleaved caspase-3 and reduced the expression of the anti-apoptotic protein Bcl-2 (Figure 6E). The above results suggest that the NCOA7 overexpression induced apoptosis and G1 phase arrest in ccRCC cells.

### 2.6. MAPK/ERK Signaling Pathway Is Involved in NCOA7-Regulated ccRCC In Vitro

To further explore the mechanism of NCOA7 in ccRCC, ccRCC data were obtained from the TCGA database and differential analysis was performed by grouping the high and low mRNA expressions of NCOA7 in tumor tissues (|log_2_fold change| ≥ 1 and *p* < 0.05). The identified differentially expressed genes were subjected to Kyoto Encyclopedia of Genes and Genomes (KEGG) enrichment analysis, and the top 30 most enriched pathways were shown in Figure 7A; these included cancer-related pathways and MAPK signaling pathways (Figure 7A). The aberrant activation of the MAPK signaling pathway is known to play an important role in the development of tumors. Consequently, we hypothesized that abnormalities in the MAPK signaling pathway were associated with the effect of NCOA7 on the malignant phenotype of ccRCC cells, and we further examined the expression levels of key proteins involved in the MAPK pathway. Western blot analysis showed reduced levels of p-ERK1/2 protein in 769-P and 786-O cells overexpressing NCOA7, whereas the levels of ERK1/2 were unaffected. In addition, there were no significant alterations in the protein levels of p38, p-p38, JNK, and p-JNK (Figure 7B). Based on these results, we infer that NCOA7 potentially regulates the proliferation, migration, and invasion of ccRCC cells through the MAPK/ERK pathway.

### 2.7. Activation of the MAPK/ERK Pathway Attenuates the Role of NCOA7

After confirming the possible involvement of NCOA7 in the regulation of the MAPK/ERK pathway, Ro 67-7476, a potent p-ERK1/2 agonist, was used to activate the activity of p-ERK1/2. The CCK8 assay, Wound healing assay, and Transwell invasion assay were used to assess the impact of the ERK1/2 activator on the proliferation, migration, and invasion of ccRCC cells. The CCK8 assay revealed that the overexpression of NCOA7 significantly inhibited the activity of 769-P and 786-O cells. Notably, the addition of the p-ERK1/2 activator significantly alleviated this inhibition (Figure 7C,D). The wound healing assay indicated that the activation of p-ERK1/2 ameliorated the inhibitory effect of NCOA7 overexpression on the migration of ccRCC cells (Figure 8A,B). The invasive ability of 769-P and 786-O cells was confirmed using Transwell invasion assays. The results demonstrated that the activation of p-ERK1/2 resisted inhibition caused by cell invasion via the overexpression of NCOA7 (Figure 8C,D). Moreover, the p-ERK1/2 activator significantly improved the inhibition of EMT caused by NCOA7 overexpression, as evidenced by the reduced protein expression of E-cadherin and increased protein expression of N-cadherin and Vimentin (Figure 8E). The downregulation of the Cyclin D1 and Bcl-2 protein expressions was significantly inhibited. The upregulation of the Bax and cleaved-caspase 3 protein expression was also inhibited (Figure 8E). These results suggest that NCOA7 inhibits the proliferation, migration, and invasion of ccRCC cells, at least in part, through the MAPK/ERK pathway.

### 2.8. NCOA7 Inhibits Tumorigenesis and Lung Metastasis of ccRCC In Vivo

The tumor-bearing model was used to assess the tumorigenic capacity of cancer cells and the effect of NCOA7 on proliferation and tumorigenesis in vivo. As shown in Figure 9A,B, 786-O cells overexpressing NCOA7 yielded smaller tumor volumes and weights than the vector group. IHC staining showed diminished Ki67 protein expression in NCOA7 overexpressing xenograft tumors compared with the vector group (Figure 9C). Consistent with the results of in vitro experiments, Western blot analysis confirmed the significant downregulation of p-ERK, Vimentin, and Bcl-2 protein expression levels in NCOA7 overexpressing allograft tumors (Figure 9D). Considering that the lung is a more common distant metastatic organ in ccRCC patients, the tail vein injection of ccRCC cells into nude mice was used to construct a lung metastasis model. Both in vitro imaging and HE assays showed that NCOA7 overexpression induced fewer nodules in lung metastases (Figure 9E,F). In conclusion, NCOA7 could inhibit the proliferation and lung metastasis of ccRCC cells in vivo to regulate the progression of ccRCC.

## 3. Discussion

The development and progression of ccRCC is a complex regulatory process, involving multiple molecular interactions regulated by key genes. Thus, studying the mechanisms regulated by these key genes may provide promising therapeutic strategies for patients with ccRCC. Previous studies have demonstrated that NCOA7 mediates the transduction of nuclear receptor signaling in specific target tissues that are mainly localized in the nucleus [14]; this is consistent with our study. Nuclear receptors play a key role in tumor suppression and promotion by recruiting nuclear receptor co-activators to enhance nuclear receptor-dependent gene transcription [20,21]. NCOA7 is found to regulate all-trans retinoic acid-mediated neuronal differentiation, and it serves as a good prognostic indicator for neuroblastoma [16]. In addition, NCOA7 exhibits a genetic polymorphism associated with the development of breast cancer, and genetic variations in its locus may reduce susceptibility to breast cancer [18,22]. The overexpression of NCOA7 promotes the proliferation of oral squamous cell carcinoma cells by activating the aryl hydrocarbon receptor (AHR) [23]. This may be related to the specificity of different tissues exhibiting different functions. A previous study confirmed that NCOA7 is highly expressed in the kidney and the involvement of NCOA7 mutations in the pathogenesis of distal renal tubular acidosis [24]. NCOA7 also senses oxidative stress and regulates the cellular response to oxidative DNA damage [25,26]. Therefore, we investigated whether NCOA7 is involved in the regulation of ccRCC and we sought to elucidate its potential function in this disease.

In this study, NCOA7 was found to be downregulated in ccRCC tissues. Bioinformatic analysis identified the prognostic significance of NCOA7, but only for ccRCC, a subtype of renal cell carcinoma. The expression levels of NCOA7 were closely related to the clinicopathological characteristics of the patients, with lower expression levels in those with high-grade and high-stage ccRCC. Patients with high NCOA7 expression levels exhibited a better prognosis compared with those with low NCOA7 expression in advanced ccRCC. These results demonstrate the significance of NCOA7 in the diagnosis and treatment of advanced or metastatic ccRCC, and thus, it warrants further study.

The dysregulation of cell proliferation, and the inhibition of apoptosis, is central to the development of all tumors; it also provides two obvious targets for therapeutic interventions in various types of cancers [27]. In this study, NCOA7 was shown to suppress the proliferation of ccRCC cells with CCK8, in vitro colony formation, and in vivo subcutaneous tumorigenesis experiments. Immunofluorescence staining also showed a significant inhibition of the proliferation marker PCNA via NCOA7 overexpression. In addition, tumor proliferation requires sustained access to survival signals to resist apoptosis [27]. Flow cytometry analysis revealed significant pro-apoptotic effects following NCOA7 overexpression, accompanied by upregulation expression levels of the apoptotic proteins, Bax and cleaved-caspase 3, and downregulation expression levels of the anti-apoptotic protein, Bcl-2. The cell cycle is a crucial driver of tumor progression and it hosts the transition between the proliferation and apoptosis of tumor cells [19]. NCOA7 overexpression induced G1 phase arrest in ccRCC cells, and the detection of cell cycle proteins revealed that the overexpression of NCOA7 significantly downregulates the protein expression of Cyclin D1, which regulates the progression from G1 to S phase [28]. In summary, NCOA7 overexpression plays an important role in the proliferation of ccRCC cells.

The uncontrolled proliferation of tumor cells eventually leads to direct invasion, lymph node metastasis, or blood metastasis, resulting in a poor prognosis. The present study revealed a negative correlation between the expression of NCOA7 and the tumor N and M stages, thus suggesting that NCOA7 inhibits the invasion and migration of ccRCC cells. Furthermore, the experiments provided evidence that the overexpression of NCOA7 can significantly reduce the invasion and migration abilities of ccRCC cells in vitro and in vivo. EMT is an important factor in tumor invasion and metastasis [29], and exhibits a strong resistance to drugs, thus making it an appealing target for therapeutic interventions in tumor treatment [30]. E-cadherin, N-cadherin, and Vimentin are the signature protein molecules of EMT. In this study, it was found that the overexpression of NCOA7 upregulated the expression levels of the E-cadherin protein and downregulated the expression levels of N-cadherin and Vimentin proteins. Therefore, NCOA7 overexpression can reduce the migration and invasion capacities of ccRCC cells, thus providing a novel biomarker and potential therapeutic target for predicting the metastasis of ccRCC cells.

According to the KEGG enrichment analysis, the MAPK signaling pathway and tumor-related signaling pathway were significantly enriched. The MAPK signaling pathway consists of at least three consecutive kinase components: MAP3K, MAP2K, and MAPK. MAP3Ks phosphorylate and activate MAP2Ks, which, in turn, phosphorylates and activates MAPKs, and activated MAPKs phosphorylate a variety of target proteins [31]. MAPKs, such as c-Jun NH2-terminal kinase (JNK), p38 MAPK, and extracellular signal-regulated kinase (ERK), are serine-threonine protein kinases that play an important role in the occurrence, progression, and drug resistance of tumors [31,32]. In the study, only the MAPK/ERK pathway was inhibited by NCOA7 overexpression, whereas the ERK1/2 activator could reduce the function of the NCOA7’s tumor suppressor gene. The RAF-MEK-ERK pathway plays a critical role in tumor proliferation and metastasis, and it is activated by upstream RAS signaling [7]. Members of the ERK signaling cascade are frequently mutated in cancer [33], and the development of specific inhibitors targeting ERK signaling members is of great significance for the treatment of ccRCC.

## 4. Materials and Methods

### 4.1. Data Sources

The RNA sequencing transcriptome dataset and clinical data of samples were downloaded from the TCGA database (https://portal.gdc.cancer.gov/, accessed on 6 September 2022) for ccRCC, kidney chromophobe (KICH), and kidney renal papillary cell carcinoma (KIRP). Moreover, GSE datasets of renal clear cell carcinoma were downloaded from the GEO database (https://www.ncbi.nlm.nih.gov/geo/, accessed on 3 May 2023) [34]. The GSE53757 dataset consists of 72 ccRCC tissues and paracancerous tissues [35]. The GSE66271 dataset comprises 13 ccRCC tissues and paracancerous tissues [36]. The GSE6344 dataset includes 10 ccRCC tissues and paracancerous tissues [37]. The GSE26574 dataset contains 8 ccRCC tissues and 8 paracancerous tissues [38]. The probe names were transformed into gene symbols based on platform annotation information.

### 4.2. Differential Expression Analysis of NCOA7

The mRNA expression differences of NCOA7 from the TCGA and the GEO database were compared in normal and tumor tissues of ccRCC patients using the R “limma” package. Data visualization was performed using the R “ggplot2” package.

### 4.3. Analysis of Prognostic Value, Diagnosis, and Clinicopathological Parameters Correlation

Survival analysis and survival curves were conducted using the R “Survival” and “survminer” packages to assess the effect of high and low expression groups of NCOA7 on overall survival (OS) and progression free survival (PFS). In addition, the R “pROC” package was used to evaluate the diagnostic accuracy of mRNA expression of NCOA7 in ccRCC and to draw receiver operating characteristic (ROC) curves. The area under the curve (AUC) of the ROC curve represents the diagnostic effect. Finally, the correlation between the mRNA expression of NCOA7 in ccRCC patients and clinicopathological features was analyzed using the Kruskal–Wallis test via the R “limma”, “ggpubr”, and “ggplot2” package.

### 4.4. Clinical Tissue Specimens

Both ccRCC tissue and matched normal kidney tissue were collected at the Renmin Hospital of Wuhan University from September 2021 to December 2022 in this study. No radiotherapy, chemotherapy, or other relevant neoadjuvant therapies were received before surgery. This project was approved by the Ethics Committee of the Renmin Hospital of Wuhan University and informed consent was obtained from all enrolled patients.

### 4.5. Cell Culture

Human ccRCC cell lines (769-P, 786-O, Caki-2, OS-RC-2, ACHN) and Human renal tubular epithelial cell lines (HK-2) were obtained from the Cell Bank of the Chinese Academy of Sciences. All cells were cultured in the DMEM-F12 medium (Gibco, Thermo Fisher Scientific, Waltham, MA, USA) supplemented with 10% fetal bovine serum (FBS; Invitrogen, Thermo Fisher Scientific, Waltham, MA, USA) and 1% antibiotics (Biosharp, Beijing, China) at 5% CO_2_ and 37 °C in a humid atmosphere.

### 4.6. Transfection of Lentiviral Plasmids

Lentiviral plasmids for the overexpression of NCOA7 and control groups were purchased from OBio Biotechnology (Shanghai, China). The Lipofectamine 2000 transfection reagent was purchased from Invitrogen (Thermo Fisher Scientific, USA). All cells (769-P, 786-O) were fused at 70% in 6-well plates within 24 h. The transfection complex was then prepared by mixing the lentivirus with 5 μL Lipofectamine 2000 transfection reagent, incubated for 15 min at room temperature, and then uniformly added to each group of cells. After 24 h of cell infection, the initial medium was replaced with a medium supplemented with 3 µg/mL puromycin. The cells were then further cultured for 2 weeks to select stable infected cell lines. Empty vectors (Vector) were used as control groups.

### 4.7. Cell Viability Assay

Cell viability assays were performed in 769-P and 786-O using the Cell Counting Kit-8 (CCK8) assay (Biosharp, Beijing, China) to investigate the inhibitory effect of NCOA7 overexpression on the proliferation of ccRCC cells. Each well of a 96-well plate was inoculated with 3 × 10^3^ cells (769-P and 786-O) and cultured for 0, 1, 2, 3, and 4 days, respectively. Cells were then incubated with a 10 ul CCK-8 solution per well for 30 min, and absorbance was measured at a wavelength of 450 nm using a Multimode Plate Reader (PerkinElmer, Waltham, MA, USA).

### 4.8. Colony Formation Assay

Colony formation assays were used to investigate the inhibitory effect of the overexpression of NCOA7 on the proliferation of ccRCC cells. The overexpression or Vector of the 769-P or 786-O cells (500 cells/well) were inoculated in 6-well plates and allowed to grow for 14 days. Subsequently, cells were fixed with 4% paraformaldehyde for 30 min and then stained with 0.1% crystal violet solution for 15 min at room temperature. The 6-well plates with colonies were air-dried at room temperature. The number of colonies was quantified to compare colony-forming abilities across different experimental conditions.

### 4.9. Wound Healing Assay

The 769-P and 786-O cells were separately seeded into 6-well plates to culture until they reached a density of 80%. The cell monolayer was scratched with 200 μL using a sterile pipette tip. Each well was washed three times with PBS and incubated with 10% FBS medium for 24 h. The wound width was imaged at 0 and 24 h using a light microscope (Olympus IX71, Japan), and the images were analyzed using ImageJ software V 1.8.0.

### 4.10. Transwell Invasion Assay

The Transwell chambers (BD Biosciences, Franklin Lakes, NJ, USA) were used to assess the effect of the overexpression of NCOA7 on the invasiveness capacity of ccRCC cells. The 769-P or 786-O cells (3 × 10^4^ cells/well) were inoculated into the upper chamber and cultured in a serum-free medium, whereas the lower chamber was filled with a complete medium containing 10% fetal bovine serum. After culturing for 24 h, the migrated cells in the lower chamber were fixed with 4% paraformaldehyde at room temperature for 30 min and stained with 0.1% crystal violet for 20 min. The migrated cells were washed with phosphate-buffered saline (PBS) (Biosharp, Beijing, China), and then observed using an inverted microscope (Olympus IX71, Japan).

### 4.11. Western Blot Assay

The total protein was extracted via the ice lysis of tumor tissue or cells with a RIPA buffer supplemented with the protease inhibitor, PMSF, for 30 min. The protein concentration of each group of samples was determined three times using the BCA method. Each group of proteins (15–30 μg) was separated using SDS-PAGE and transferred into PVDF membranes (Millipore, NJ, USA). After being blocked with 5% non-fat-milk in Tris-buggered saline with 0.1% Tween-20 (TBS-T) for 1.5 h at room temperature, membranes were incubated overnight with the corresponding primary antibody at 4 °C: anti-β-actin (1:1000, 23660-1-AP, Proteintech Group); anti-NCOA7 (1:1000, 23092-1-AP, Proteintech Group); anti-Bax (1:1000, 50599-2-Ig, Proteintech Group); anti-Bcl-2 (1:1000, 4223, Cell Signaling Technology); anti-cleaved caspase-3 (1:1000, 9661, Cell Signaling Technology); anti-Cyclin D1 (1:1000, 26939-1-AP, Proteintech Group); anti-N-cadherin (1:1000, 22018-1-AP, Proteintech Group); anti-E-cadherin (1:1000, 20874-1-AP, Proteintech Group); anti-Vimentin (1:1000, 10366-1-AP, Proteintech Group); anti-ERK1/2 (1:1000, 4695, Cell Signaling Technology); anti-phospho-ERK1/2 (1:1000, 4370, Cell Signaling Technology); anti-JNK (1:1000, 9252, Cell Signaling Technology); anti-phospho-JNK (1:1000, 4668, Cell Signaling Technology); anti-p38 (1:1000, 8690, Cell Signaling Technology); and anti-phospho-p38 (1:1000, 4511, Cell Signaling Technology). After washing, the membrane was further incubated with the enzyme-labeled secondary antibody (1:10,000) for 1 h at room temperature the next day and visualized with an enhanced chemiluminescence analysis kit (Biosharp, Beijing China) on the ChemiDoc MP imaging system (Bio-rad, Hercules, CA, USA). β-actin was used as the internal loading control and all target proteins were analyzed with the ImageJ software.

### 4.12. Immunofluorescence Staining

Slides containing 769-P or 786-O cells were fixed in 4% paraformaldehyde with bovine serum albumin, primary antibody PCNA (1:100, CL594-10205, Proteintech Group) or Vimentin (1:100, 10366-1-AP, Proteintech Group), and E-cadherin (1:100, 20874-1-AP, Proteintech Group) overnight at 4 °C. The following day, slices were incubated with corresponding fluorescent labeled secondary antibodies at room temperature for 1 h. Cell nuclei were stained with DAPI for 10 min, changes in protein expression were observed via fluorescent microscopy (Olympus BX51, Tokyo, Japan), and image acquisition was performed.

### 4.13. Histology and Immunohistochemistry Assay

Tumor tissue sections of 5 μm thicknesses were taken, then, they were dewaxed in xylene at room temperature for 20 min, rehydrated in ethanol, and stained with hematoxylin and eosin for histological examination (HE).

Slices were placed in an EDTA buffer for microwave repair and treated with 3% hydrogen peroxide for 10 min to inhibit endogenous peroxidase activity. Slides were incubated with 5% BSA for 20 min and with 5% BSA for 20 min. The tissue slices were then left overnight at 4 °C with the primary antibody. Slices were washed three times with PBS and exposed to the appropriate secondary antibodies for 1 h at room temperature. DAB was used to develop the color for 5 min and hematoxylin was used to retain the nuclei for 5 min. An Olympus BX51microscope (Olympus, Japan) was used to visualize the images. Ten randomly selected microscopic areas per slide were observed by two independent pathologists; the degree of positive staining was recorded. The IHC score was calculated in accordance with the intensity and extent of the staining (staining intensity: negative = 0, weak = 1, moderate = 2, strong = 3; staining degree: 0 = no staining, 1 = 0–10%, 2 = 10–50%, and 3 = 50–100%). The total score was calculated by multiplying the intensity score with the degree score. Scores ranging from 0 to 3 were considered to be a negative expression, scores from 4 to 6 indicate weak expression, and scores from 8 to 12 represent strong expression.

### 4.14. Flow Cytometry

Cells were digested into a single cell suspension with 0.25% tryp-sin (without EDTA), and approximately 6 × 10^5^ cells were collected via centrifugation at 1000 RPM. After washing twice with PBS, the staining solution was added to all samples for 30 min of incubation in the dark. Cell fluorescence signals were detected using a CytoFLEX flow cytometer (Beckman Coulter, Brea, CA, USA) to determine cell cycle phase distribution, and the results were analyzed using Modfit LT 5.0 software.

An Annexin V-FITC/PI Cell Apoptosis Detection Kit (Servicebio, Wuhan, China) was used to detect the apoptosis rate of the ccRCC cells. The 769-P or 786-O cells (10,000 cells/mL) were plated in a 6-well plate and cultured overnight. In accordance with the manufacturer’s instructions, the cells were collected via centrifugation after digestion with trypsin without EDTA. After washing twice with PBS, the cells were suspended in a 400 μL Annexin V conjugate; then, 5 μL Annexin V-FITC was added and left to stand for 15 min protected from light, and 10 μL PI was added and left to stand for 10 min protected from light. CytoFLEX flow cytometry (Beckman Coulter) was used for detection immediately after staining. In the four quadrants, the cells in the lower left quadrant were live cells, the cells in the upper right quadrant were late apoptotic cells, the cells in the lower right quadrant were early apoptotic cells, the cells in the upper left quadrant were necrotic cells. The apoptosis rate was calculated using the sum of the number of apoptotic cells in the upper and lower right quadrants for further statistical analysis.

### 4.15. In Vivo Assays

Animal experiments were approved by the Ethical Committee for Animal Experiments of the Renmin Hospital of Wuhan University. All procedures involving mice were performed in accordance with the National Institutes of Health Guide for the Care and Use of Laboratory Animals.

A tumor-bearing model (four mice per group) was established by subcutaneously injecting 5 × 10^6^/100 μL cells into the right side of four-week-old male nude mice (Hunan Slike Jingda Laboratory Animals, Changsha, China). Tumor growth was monitored every 5 days for a duration of 1 month. Subsequently, the mice were euthanized, and the tumors were dissected, photographed and weighed. Tumor tissue was preserved in either liquid nitrogen or formaldehyde for further experiments.

A lung metastasis model (three mice per group) was established via a tail vein injection of 10^6^/100 μL of cells into nude mice. Two weeks later, the mice were injected intraperitoneally with luciferase at a concentration of 15 mg/mL and subsequently euthanized after 15 min. The lung tissues were removed and placed on the darkroom imaging platform IVIS Lumina III to assess the extent and distribution of metastases via luminescence of the lungs. Finally, the lung tissue was preserved in formaldehyde for subsequent experiments.

### 4.16. Statistical Analysis

All experiments were performed at least three times and data were expressed as mean ± SD, and they were calculated and statistically analyzed using SPSS 26.0 statistical software and Microsoft Excel 2016. *p*-values expressing significant *p*-values for differences were calculated using the Student’s *t*-test for a comparison of two groups. Comparisons between more than two groups were statistically analyzed using one-way ANOVA, and Tukey’ post hoc test was used for pairwise comparisons after one-way ANOVA. Differences with *p*-values < 0.05 were considered statistically significant. Graphs were constructed using GraphPad Prism 8.0 software.

## 5. Conclusions

Taken together, the findings of this study suggest that NCOA7 regulates the proliferation, invasion, and migration of ccRCC cells through the MAPK/ERK pathway. NCOA7 is a promising biomarker for predicting prognosis in patients with ccRCC, especially those with advanced disease, and this study may provide new insights into therapeutic strategies for ccRCC.

## Figures and Tables

**Figure 1 ijms-24-11584-f001:**
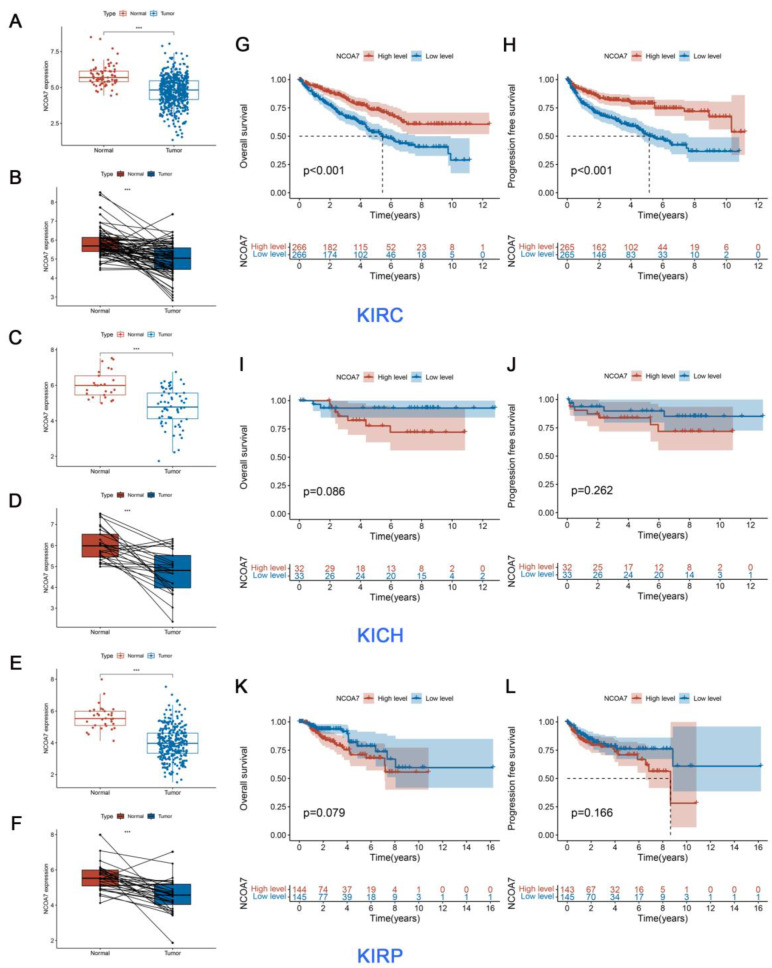
NCOA7 expression levels and its prognostic roles in three subtypes of RCC. (**A**,**C**,**E**) The different expression levels of NCOA7 in the three subtypes of RCC tissues and adjacent normal tissues. (**B**,**D**,**F**) The different expression levels of NCOA7 in the three subtypes of RCC tissues and paired normal tissues. (**G**,**I**,**K**) Overall survival curve of three subtypes of RCC patients with low and high NCOA7 expression in the TCGA database. (**H**,**J**,**L**) Progress free survival curve of three subtypes of RCC patients with low and high NCOA7 expression in the TCGA database. KIRC, kidney renal clear cell carcinoma; KICH, kidney Chromophobe; KIRP, kidney renal papillary cell carcinoma. *** *p* < 0.001.

**Figure 2 ijms-24-11584-f002:**
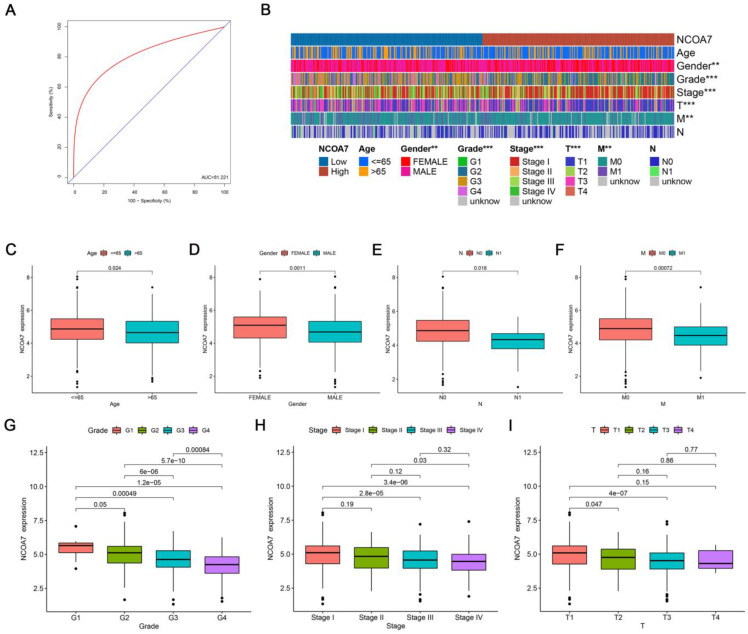
Clinicopathologic characteristics and diagnosis of the NCOA7 expression levels of ccRCC. (**A**) The ROC curve showed the efficiency of NCOA7 expression level in distinguishing ccRCC tissues from non-tumor tissues. (**B**) Heatmap of clinicopathologic features of NCOA7 in high and low expression groups. (**C**–**I**) Relative expression levels of NCOA7 in the ccRCC of the TCGA database with different ages (**C**), genders (**D**), N stages (**E**), M stages (**F**), grades (**G**), stages (**H**), and T stages (**I**). ** *p* < 0.01, *** *p* < 0.001.

**Figure 3 ijms-24-11584-f003:**
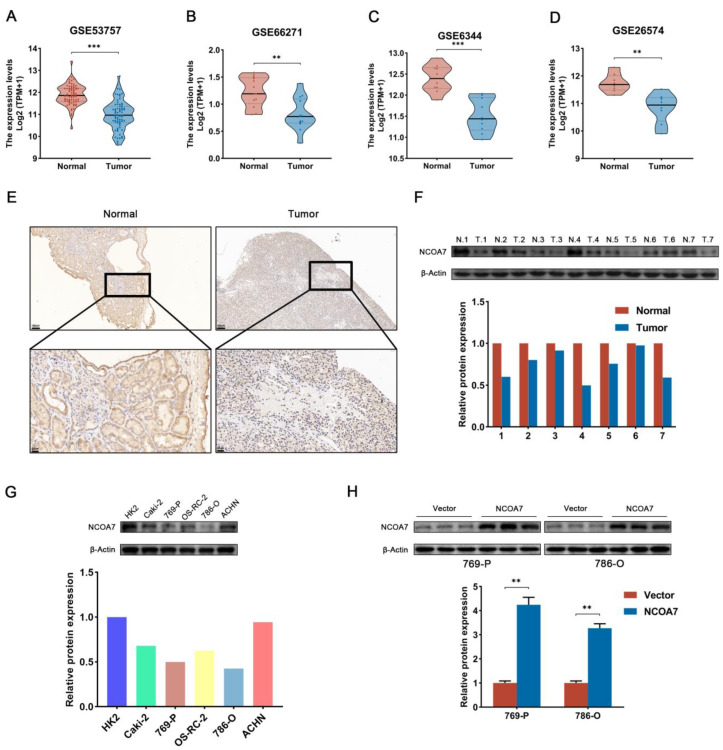
Expression of NCOA7 in ccRCC tissues and cell lines. (**A**–**D**) The expression levels of NCOA7 in ccRCC tissues and adjacent normal tissues in GSE53757 (**A**), GSE66271 (**B**), GSE6344 (**C**), and GSE26574 (**D**). (**E**) Representative images of different expression levels of NCOA7 in ccRCC tissues and adjacent normal tissues via IHC staining. (**F**) Western blot analysis of NCOA7 protein expression in seven randomly selected paired specimens of ccRCC. (**G**) The protein expression levels of NCOA7 in five ccRCC cell lines. (**H**) NCOA7 overexpression efficiency was determined via Western blot analysis in 769-P and 786-O. Scale bars = 100 μm or 20 μm. ** *p* < 0.01, *** *p* < 0.001.

**Figure 4 ijms-24-11584-f004:**
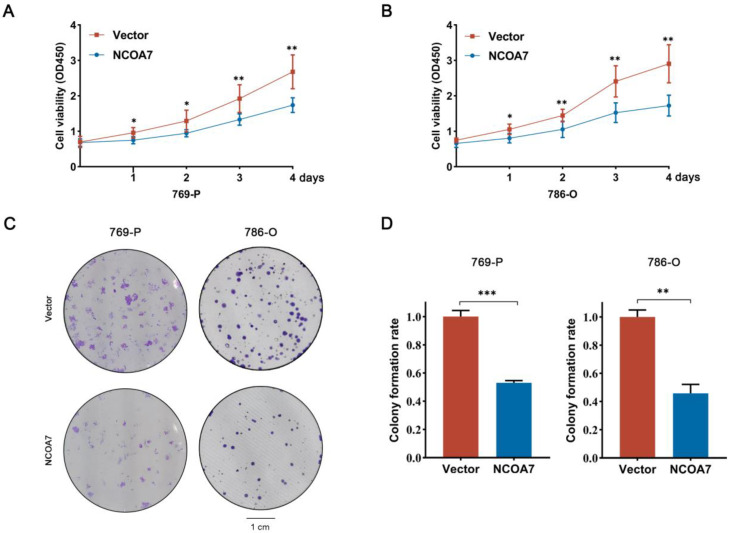
Overexpression of NCOA7 inhibits ccRCC cell proliferation in vitro. (**A**,**B**) Cell proliferation in ccRCC cells with different NCOA7 expression levels was assessed using CCK-8 assays. (**C**,**D**) Colony formation assays were performed in ccRCC cells with different NCOA7 expression levels. (**E**) Effects of NCOA7 overexpression on cell proliferation using PCNA immunofluorescence staining assays. Scale bars = 20 μm. * *p* < 0.05, ** *p* < 0.01, *** *p* < 0.001.

**Figure 5 ijms-24-11584-f005:**
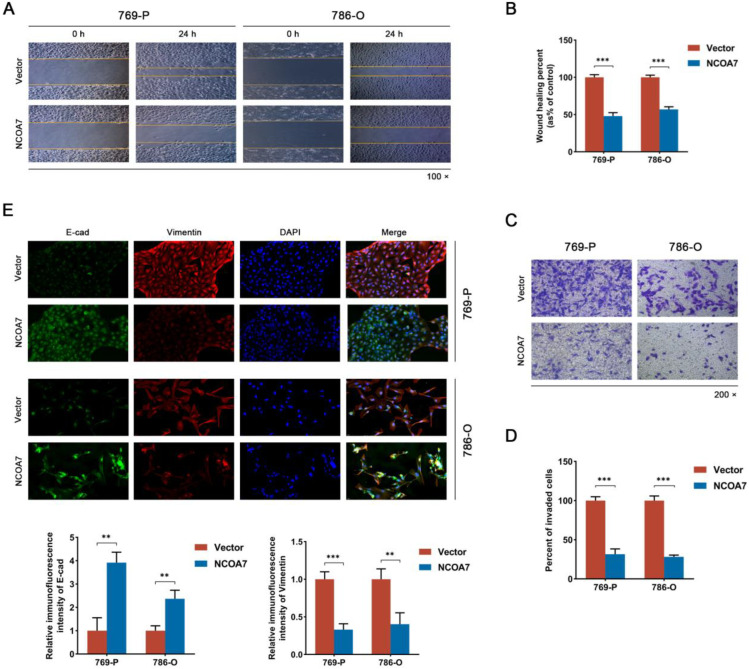
Overexpression of NCOA7 inhibits migration, invasion, and EMT of ccRCC cells in vitro. (**A**,**B**) Wound healing assays were performed to evaluate the migration of ccRCC cells (**A**) and the percentage of wound closure was calculated (**B**); scale bar 100 μm. (**C**,**D**) Transwell assays showed the effect of NCOA7 overexpression on migration and the invasive abilities of ccRCC cells; scale bar 50 μm. (**E**) Representative fluorescence images of Vimentin and E-cadherin in 769-P and 786-O cells; scale bar 50 μm. (**F**) Western blot was used to detect changes in EMT-related proteins in 769-P and 786-O cells. N-cad, N-cadherin; E-cad, E-cadherin. * *p* < 0.05, ** *p* < 0.01, *** *p* < 0.001.

**Figure 6 ijms-24-11584-f006:**
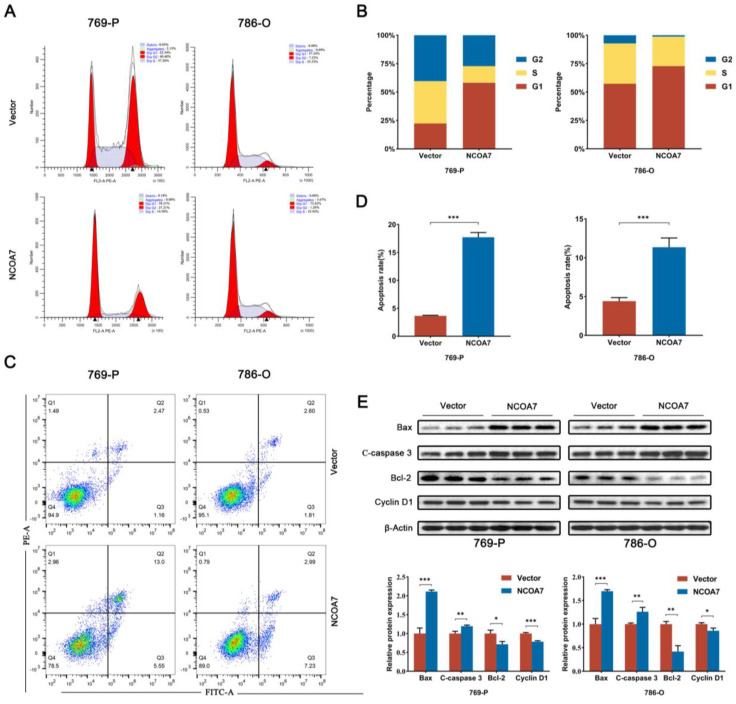
Overexpression of NCOA7 inhibits the cell cycle and apoptosis in ccRCC cells. (**A**,**B**) Effect of NCOA7 overexpression on the cell cycle via flow cytometry assay in 769-P and 786-O cells. (**C**,**D**) Effect of NCOA7 overexpression on cell apoptosis, in accordance with the flow cytometry analysis, in ccRCC cells. (**E**) The expressions of Bax, Bcl-2, Cleaved caspase-3, and Cyclin D1 in 769-P and 786-O cells were analyzed using Western blot. * *p* < 0.05, ** *p* < 0.01, *** *p* < 0.001.

**Figure 7 ijms-24-11584-f007:**
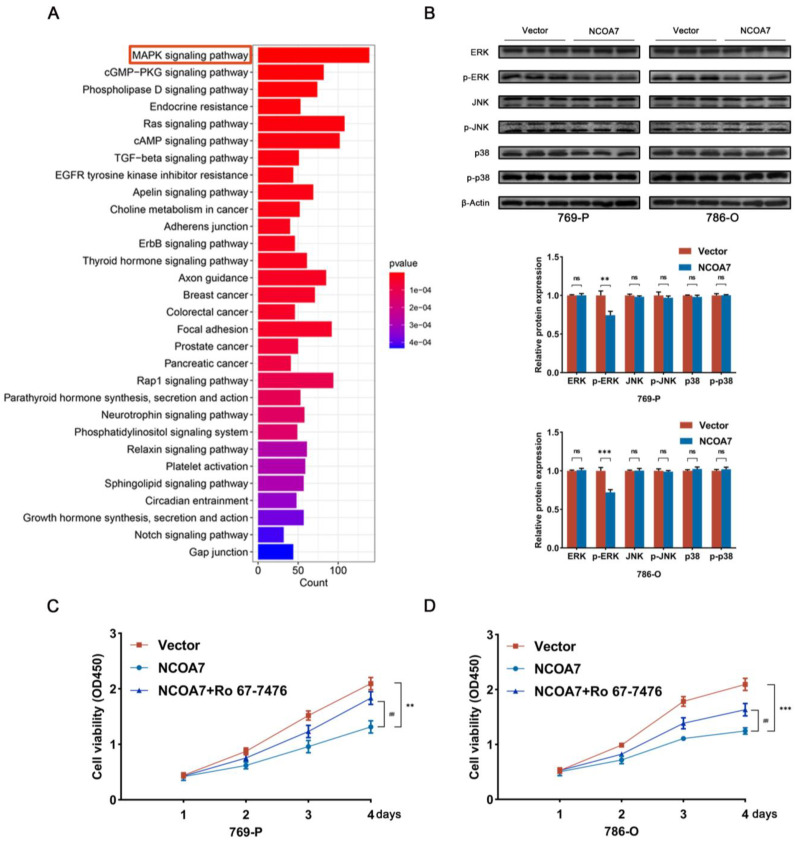
Overexpression of NCOA7 inhibits the MAPK signaling pathway in vitro. (**A**) The analysis of the enrichment results of the KEGG pathway regarding the high and low expression of NCOA7. (**B**) The inhibition effect of NCOA7 overexpression on the MAPK signaling pathway proteins were detected using Western blot in 769-P and 786-O cells. (**C**,**D**) The effects of Ro 67-7476 on NCOA7-inhibited cell proliferation were investigated using CCK-8 assays in ccRCC cells. Ro 67-7476 was added as an p-ERK1/2 activator. ** *p* < 0.01 vs. control group, *** *p* < 0.001 vs. control group, ## *p* < 0.01 vs. NCOA7 overexpression + Ro 67-7476 group. ns, means no significant.

**Figure 8 ijms-24-11584-f008:**
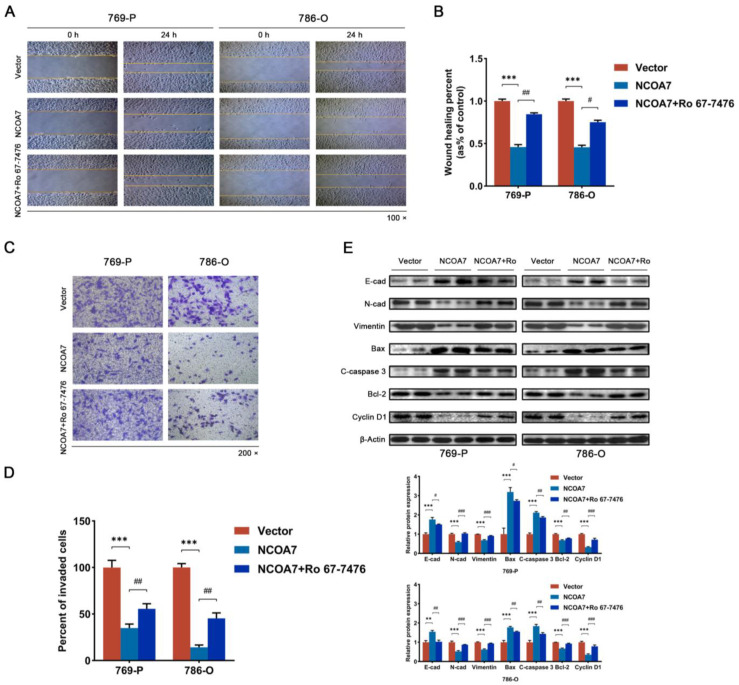
NCOA7 inhibits migration, invasion, and EMT by regulating MAPK/ERK signaling pathways in vitro. (**A**,**B**) The p-ERK activator Ro 67-7476 partially restores the inhibitory effect of NCOA7 overexpression on cell migratory capacity using wound healing assays; quantitative analysis of cell migratory capacity; scale bar 100 μm. (**C**,**D**) The p-ERK activator, Ro 67-7476, partially restores the inhibitory effect of NCOA7 overexpression on cell migration and invasion capacity using Transwell assays; quantitative analysis of cell migration capacity; scale bar 50 μm. (**E**) NCOA7 affected EMT and the cell cycle via the regulation of the MAPK/ERK signaling pathway using Western blot analysis. E-cad, E-cadherin; N-cad, N-cadherin. ** *p* < 0.01 vs. control group, *** *p* < 0.001 vs. control group, # *p* < 0.01 vs. NCOA7 overexpression + Ro 67-7476 group, ## *p* < 0.01 vs. NCOA7 overexpression + Ro 67-7476 group, ### *p* < 0.01 vs. NCOA7 overexpression + Ro 67-7476 group.

**Figure 9 ijms-24-11584-f009:**
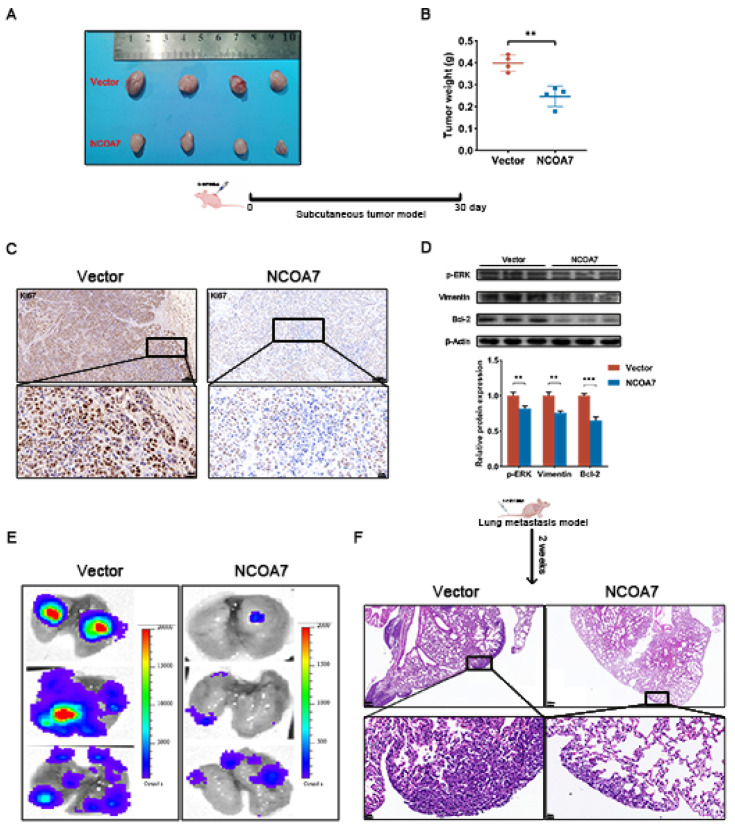
NCOA7 inhibits ccRCC cells growth and metastasis in vivo. (**A**,**B**) Effect of NCOA7 overexpression on ccRCC tumorigenesis in vivo. The weight of subcutaneous tumors was measured (n = 4). (**C**) IHC staining of Ki67 in nude mouse xenograft tumors derived from 786-O-NCOA7 overexpression cells and controls; scale bar 100 μm. (**D**) Changes in p-ERK, Vimentin, and Bcl-2 proteins were detected via Western blot in vivo. (**E**) Fluorescence imaging using a live imager to observe the effect of NCOA7 overexpression on lung metastasis of ccRCC cells. (**F**) Representative HE images of lung metastasis; scale bar 200 μm. ** *p* < 0.01, *** *p* < 0.001.

## Data Availability

The transcriptome sequencing data of renal cell carcinoma analyzed during the current study are available in TCGA database (https://portal.gdc.cancer.gov/, accessed on 6 September 2022). Four datasets (GSE53757, GSE66271, GSE6344, and GSE26574) analyzed during the present study are available in the Gene Expression Omnibus (https://www.ncbi.nlm.nih.gov/geo/, accessed on 3 May 2023).

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
