# Peer review of "NCOA7 Regulates Growth and Metastasis of Clear Cell Renal Cell Carcinoma via MAPK/ERK Signaling Pathway"

_ijms, 2023, doi:10.3390/ijms241411584_

Round 1

Reviewer 1 Report

Jiayu Guo and coauthors described the possible new regulatory protein in progression of clear cell renal carcinoma. They provided mechanistical and statistical evidence on role of NCOA7 in the inhibition of renal cancer proliferation and metastases. The paper is written properly, adequately concise and informative, data are followed by statistical analysis. They used correct methodology and various technics including bioinformatic analysis of deta deposited in database, NCOA7 overexpression, pERK antagonist treatment, cellular and molecular technics. Importantly, results support the conclusion. Therefore, my only thought is language, which requires correction in many places.

Minor revision

The manuscript should be proofread by a native speaker.

Reviewer 2 Report

This manuscript deals with the role of the NCOA7 molecule in regulating the cell growth and Metastasis of Clear Cell Renal Cell Carcinoma  (ccRCC). This regulation is dependent on the MAPK/ERK signaling pathway.

Overall, the manuscript is of interest and the analysis is novel.

There are some major concerns that should be fully addressed to endorse the manuscript for publication.

Some notes are enclosed along the manuscript and the authors should reply to all these matters.

Briefly, the experiments on overexpressing ccRCC lines are of interest. However, it is not shown if the cell populations overexpressing the NCOA7 are homogeneously overexpressing this molecule or not. No information is reported on the efficiency, selection, and actual expression level of the molecule considered. These experiments can be easily shown using immunofluorescence assays. I think also that a homogeneous cell populations for the molecule of interest should be used, but the authors did not mention any selection of transfected clones and so on.

It is essential that the purity and features of the cell lines used after the overexpression are shown to interpret the findings reported.

Minor point

The figures are not embedded in the manuscript as usual. Please follow the rules for the submission of the revised version.

I think that the manuscript is enough well written.

Round 2

Reviewer 2 Report

There are some misprints, mainly in the new red portion of the manuscript. Please check it.

I think English language is good.

Author Response

Dear reviewer,

Thank you for your reviewing. We have corrected manuscript. If there is anything else you would like to change, please do not hesitate to contact us!